# Beyond ReinMax: Low-Variance Gradient Estimators for Discrete Latent Variables

**Daniel Wang**  *u7918232@anu.edu.au*
*School of Computing*
*College of Systems and Society*
*Australian National University*

**Thang D. Bui**  *thang.bui@anu.edu.au*
*School of Computing*
*College of Systems and Society*
*Australian National University*

**Reviewed on OpenReview:** *https://openreview.net/forum?id=crlvtnsyIT*

## Abstract

Machine learning models involving discrete latent variables require gradient estimators to facilitate backpropagation in a computationally efficient manner. The most recent addition to the Straight-Through family of estimators, ReinMax, can be viewed from a numerical ODE perspective as incorporating an approximation via Heun's method to reduce bias, but at the cost of high variance. In this work, we introduce the ReinMax-Rao and ReinMax-CV estimators which incorporate Rao-Blackwellisation and control variate techniques into ReinMax to reduce its variance. Our estimators demonstrate superior performance on training variational autoencoders with discrete latent spaces. Furthermore, we investigate the possibility of leveraging alternative numerical methods for constructing more accurate gradient approximations and present an alternative view of ReinMax from a simpler numerical integration perspective.

## 1 Introduction

In machine learning problems which involve optimising the parameters of a discrete categorical distribution, gradient-based optimisation methods relying on backpropagation require computing derivatives with respect to the non-differentiable operation of sampling a random variable from a discrete distribution. In practical settings, this is typically done via estimators of an exact expression for the gradient that is computationally expensive to evaluate.

These estimators are based on either the REINFORCE estimator (Williams, 1992) or the Straight-Through estimator (Bengio et al., 2013). REINFORCE-style estimators are unbiased but suffer from high variance which limits their applicability in high-dimensional settings, although recent work by Drolet et al. (2026) has demonstrated competitive performance on discrete variational auto-encoders. On the other hand, the Straight-Through family of estimators have lower variance but at the cost of being biased. In this paper, we focus on the family of Straight-Through gradient estimators. In its simplest form, the Straight-Through estimator (Bengio et al., 2013) is constructed by heuristically approximating the Jacobian of the non-differentiable sampling of a discrete categorical random variable as the identity function. Improvements to the Straight-Through estimator in recent years have leveraged various mathematical tools. Based on the Gumbel-Softmax reparameterisation (Jang et al., 2017; Maddison et al., 2017), the Straight-Through Gumbel-Softmax estimator is constructed by differentiating through a continuous relaxation of the sampling of a categorical random variable. The Gumbel-Rao estimator extends this by making use of the known reparameterisation of the conditional distribution to reduce variance via conditional marginalisation

(Paulus et al., 2021). Proceeding in a numerical ODE direction, it can be shown that the Straight-Through estimator can be viewed as incorporating a first-order Forward Euler approximation (Liu et al., 2023). Extending this to incorporate the more accurate second-order Heun's method leads to the ReinMax estimator which has much lower bias at the cost of higher variance.

To remedy this high variance, we combine the Gumbel-Softmax reparameterisation and numerical ODE approaches by incorporating Gumbel-Softmax tricks into the ReinMax estimator, firstly via a heuristic Gumbel-Rao approximation and secondly via control variates to arrive at our ReinMax-Rao and ReinMax-CV estimators. Extensive experiments on training variational autoencoders (VAEs) (Kingma & Welling, 2014) with discrete latent spaces show that our methods outperform ReinMax and provide insight into the bias-variance trade-off for these gradient estimators.

Although the main contribution of this paper is variance reduction of ReinMax, it is also worth investigating possible approaches to further reduce its bias by leveraging alternative numerical methods. To this end, we generalise the construction of ReinMax from Heun's method to the entire family of second-order Runge-Kutta ODE methods. However, this is ineffective in practice and we argue that a different set of numerical integration methods are more appropriate for this problem.

## 2 Background

Consider an $n$-class discrete random variable $\boldsymbol{D} \in \{\boldsymbol{I}_1, \boldsymbol{I}_2, ..., \boldsymbol{I}_n\}$ represented as a one-hot encoding (where $\boldsymbol{I}_i$ is the $i^{\text{th}}$ standard basis vector in $\mathbb{R}^n$), distributed according to $\boldsymbol{D} \sim \boldsymbol{\pi} = \text{softmax}(\boldsymbol{\theta})$ where $\boldsymbol{\theta} \in \mathbb{R}^n$ are optimisable parameters. Given a differentiable loss function $f : \mathbb{R}^n \to \mathbb{R}$ that takes as input a realisation of $\boldsymbol{D}$, the objective is to minimise the expected loss w.r.t. $\boldsymbol{\theta}$:

$$\widehat{\boldsymbol{\theta}} = \arg\min_{\boldsymbol{\theta}} \mathbb{E}_{\boldsymbol{D} \sim \text{softmax}(\boldsymbol{\theta})}[f(\boldsymbol{D})]. \tag{1}$$

The derivative with respect to $\boldsymbol{\theta}$ is given by

$$\nabla := \frac{\partial}{\partial \boldsymbol{\theta}} E_{\boldsymbol{D} \sim \text{softmax}(\boldsymbol{\theta})}[f(\boldsymbol{D})] = \frac{\partial}{\partial \boldsymbol{\theta}} \sum_{i=1}^{n} \boldsymbol{\pi}_i f(\boldsymbol{I}_i) = \sum_{i=1}^{n} f(\boldsymbol{I}_i) \frac{d\boldsymbol{\pi}_i}{d\boldsymbol{\theta}}. \tag{2}$$

This expression is expensive to compute because it requires $n$ evaluations of $f$ which is impractical in modern deep learning settings where $f$ is a neural network. Hence, estimators of the gradient that require only one forward and backward pass are desired.

### 2.1 The Straight-Through estimator (Bengio et al., 2013)

The Straight-Through estimator (Bengio et al., 2013) preserves the non-differentiable sampling of $\boldsymbol{D}$ in the forward pass while replacing it with a differentiable approximation in the backward pass. Specifically, given a realisation of the categorical random variable $\boldsymbol{D} \sim \boldsymbol{\pi} := \text{softmax}(\boldsymbol{\theta})$, the Straight-Through estimator is given by

$$\widehat{\nabla}_{\text{ST},\tau}(\boldsymbol{D}, \boldsymbol{\theta}) := \frac{\partial f(\boldsymbol{D})}{\partial \boldsymbol{D}} \cdot \frac{d\boldsymbol{\pi}_\tau}{d\boldsymbol{\theta}} = \frac{\partial f(\boldsymbol{D})}{\partial \boldsymbol{D}} (\text{diag}(\boldsymbol{\pi}_\tau) - \boldsymbol{\pi}_\tau \boldsymbol{\pi}_\tau^\top), \ \ \boldsymbol{\pi}_\tau := \text{softmax}_\tau(\boldsymbol{\theta}) \tag{3}$$

where $\text{softmax}_\tau(\boldsymbol{\theta}) := \text{softmax}(\frac{\boldsymbol{\theta}}{\tau})$ is the tempered softmax function and $\tau$ is a hyperparameter referred to as the temperature that is typically set to $\tau = 1$ for this estimator. The Straight-Through estimator can be interpreted as simply approximating the Jacobian of the non-differentiable operation of sampling $\boldsymbol{D}$ by the identity matrix (i.e., $\frac{\partial \boldsymbol{\pi}}{\partial \boldsymbol{D}} \approx \boldsymbol{I}$) in the backward pass, and is therefore biased.

### 2.2 The Straight-Through Gumbel-Softmax estimator (Jang et al., 2017)

Motivated by the reparameterisation trick that is commonly used for continuous random variables, Jang et al. (2017) and Maddison et al. (2017) introduced the Gumbel-Softmax reparameterisation for sampling $\boldsymbol{D} \sim \boldsymbol{\pi} = \text{softmax}(\boldsymbol{\theta})$:

$$\boldsymbol{D} \overset{d}{=} \lim_{\tau \to 0} \text{softmax}_\tau(\boldsymbol{\theta} + \boldsymbol{G}) \tag{4}$$

where $G$ is a vector of i.i.d. Gumbel$(0, 1)$ random variables. Note that the zero-temperature limit of the softmax$(\cdot)$ is the one-hot embedding of the $\arg\max(\cdot)$ function, which is non-differentiable. Using this reparameterisation, the Straight-Through Gumbel-Softmax estimator is given by

$$\widehat{\nabla}_{\text{STGS},\tau}(\boldsymbol{G}, \boldsymbol{\theta}) := \frac{\partial f(\boldsymbol{D})}{\partial \boldsymbol{D}} \cdot \frac{d\,\text{softmax}_\tau(\boldsymbol{\theta} + \boldsymbol{G})}{d\boldsymbol{\theta}}. \tag{5}$$

In the forward pass, the one-hot vector $\boldsymbol{D}$ is obtained by taking the zero-temperature limit of softmax$(\cdot)$ as in Equation 4 while in the backward pass, a non-zero (typically small) temperature is chosen so that $\frac{d\,\text{softmax}_\tau(\boldsymbol{\theta}+\boldsymbol{G})}{d\boldsymbol{\theta}}$ is well-defined. In practice, choosing a smaller value of $\tau$ leads to low bias but high variance.

### 2.3 The Gumbel-Rao estimator (Paulus et al., 2021)

While the Gumbel-Softmax reparameterisation samples $\boldsymbol{G}$ first then obtains $\boldsymbol{D}$ as the deterministic $\arg\max(\cdot)$ of $\boldsymbol{\theta} + \boldsymbol{G}$, Paulus et al. (2021) propose to sample $\boldsymbol{D} \sim \text{softmax}(\boldsymbol{\theta})$ first, then $\boldsymbol{\theta} + \boldsymbol{G}$ conditional on $\boldsymbol{D}$. The Gumbel-Rao estimator makes use of the conditional marginalisation $\mathbb{E}_{\boldsymbol{G}}\left[\widehat{\nabla}_{\text{STGS},\tau}(\boldsymbol{G}, \boldsymbol{\theta})\right]$ $= \mathbb{E}_{\boldsymbol{D}}\left[\mathbb{E}_{\boldsymbol{\theta}+\boldsymbol{G}|\boldsymbol{D}}\left[\widehat{\nabla}_{\text{STGS},\tau}(\boldsymbol{G}, \boldsymbol{\theta})\right]\right]$ where the Rao-Blackwell Theorem (Blackwell, 1947) implies that this estimator will have the same expectation as Straight-Through Gumbel-Softmax but lower variance. More precisely, the Gumbel-Rao estimator is given by

$$\frac{\partial f(\boldsymbol{D})}{\partial \boldsymbol{D}} \mathbb{E}_{\boldsymbol{\theta}+\boldsymbol{G}|\boldsymbol{D}}\left[\frac{d\,\text{softmax}_\tau(\boldsymbol{\theta} + \boldsymbol{G})}{d\boldsymbol{\theta}}\right] \approx \frac{\partial f(\boldsymbol{D})}{\partial \boldsymbol{D}}\left[\frac{1}{K}\sum_{k=1}^{K}\frac{d\,\text{softmax}_\tau(\boldsymbol{Y}_{\boldsymbol{\theta},\boldsymbol{D},k})}{d\boldsymbol{\theta}}\right] =: \widehat{\nabla}_{\text{GR},\tau}(\boldsymbol{D}, \boldsymbol{\theta}) \tag{6}$$

where the expectation is computed via a Monte-Carlo approximation with $K$ samples of the random variable $\boldsymbol{Y}_{\boldsymbol{\theta},\boldsymbol{D}} := \boldsymbol{\theta} + \boldsymbol{G} \mid \boldsymbol{D}$. Note that this Monte-Carlo approximation means that the Gumbel-Rao estimator (and our proposed estimators which also use the Monte-Carlo approximation) is generally around two to three times slower than the other estimators. Crucially, sampling from this conditional distribution is made possible by the following reparameterisation (Maddison et al., 2014; Maddison, 2016; Tucker et al., 2017)

$$\boldsymbol{\theta}_j + \boldsymbol{G}_j \mid (\boldsymbol{D} = \boldsymbol{I}_i) \ \overset{d}{=} \ \begin{cases} -\log(E_j) + \log Z(\boldsymbol{\theta}), & \text{if } j = i, \\ -\log\left(\dfrac{E_j}{\exp(\boldsymbol{\theta}_j)} + \dfrac{E_i}{Z(\boldsymbol{\theta})}\right), & \text{otherwise} \end{cases} \tag{7}$$

where $E_j \sim \exp(1)$ i.i.d. and $Z(\boldsymbol{\theta}) = \sum_{i=1}^{n}\exp(\theta_i)$.

It is important to note that in practice, the implementation of Gumbel-Rao differs from Equation 6. Specifically, the $\frac{d\,\text{softmax}_\tau(\boldsymbol{Y}_{\boldsymbol{\theta},\boldsymbol{D},k})}{d\boldsymbol{\theta}}$ term is implemented as $\frac{d\,\text{softmax}_\tau(\boldsymbol{Y}_{\boldsymbol{\theta},\boldsymbol{D},k})}{d\boldsymbol{Y}_{\boldsymbol{\theta},\boldsymbol{D},k}}$ which ignores the derivative through the conditional reparameterisation. It is unclear from the Paulus et al. (2021) paper whether this is the intended behaviour, but it is the approach taken by the current publicly available implementations (nshepperd, 2021; Fan et al., 2022) and works well in practice so we follow it for our estimators.

### 2.4 The ReinMax estimator (Liu et al., 2023)

Returning to Equation 2, Liu et al. (2023) show that by considering baseline subtraction with the baseline $E_{\boldsymbol{D}\sim\text{softmax}(\boldsymbol{\theta})}[f(\boldsymbol{D})]$, it can be equivalently expressed as

$$\nabla = \sum_{i=1}^{n}\sum_{j=1}^{n}\boldsymbol{\pi}_j(f(\boldsymbol{I}_i) - f(\boldsymbol{I}_j))\frac{d\boldsymbol{\pi}_i}{d\boldsymbol{\theta}}. \tag{8}$$

From this expression, Equation 8 is manipulated into a more convenient form by approximating $f(\boldsymbol{I}_i) - f(\boldsymbol{I}_j)$ in terms of $\frac{\partial f(\boldsymbol{I}_i)}{\partial \boldsymbol{I}_i}$ and $\frac{\partial f(\boldsymbol{I}_j)}{\partial \boldsymbol{I}_j}$. Specifically, by replacing $f(\boldsymbol{I}_i) - f(\boldsymbol{I}_j)$ in Equation 8 with $\frac{\partial f(\boldsymbol{I}_j)}{\partial \boldsymbol{I}_j}(\boldsymbol{I}_i - \boldsymbol{I}_j)$ which

can be interpreted as a first-order Forward Euler method approximation, we have

$$\widehat{\nabla}_{\text{1st-order}} := \sum_{i=1}^{n} \sum_{j=1}^{n} \boldsymbol{\pi}_j \frac{\partial f(\boldsymbol{I}_j)}{\partial \boldsymbol{I}_j} (\boldsymbol{I}_i - \boldsymbol{I}_j) \frac{d\boldsymbol{\pi}_i}{d\boldsymbol{\theta}}. \tag{9}$$

Liu et al. (2023) showed that the Straight-Through estimator with $\tau = 1$ is equal to the first-order approximation in Equation 9 in expectation. That is,

$$E_{\boldsymbol{D}\sim\boldsymbol{\pi}}[\widehat{\nabla}_{\text{ST}}(\boldsymbol{D}, \boldsymbol{\theta})] = \widehat{\nabla}_{\text{1st-order}}. \tag{10}$$

Based on this, a more accurate second-order Heun's method approximation which replaces $f(\boldsymbol{I}_i) - f(\boldsymbol{I}_j)$ in Equation 8 with $\frac{1}{2}(\frac{\partial f(\boldsymbol{I}_i)}{\partial \boldsymbol{I}_i} + \frac{\partial f(\boldsymbol{I}_j)}{\partial \boldsymbol{I}_j})(\boldsymbol{I}_i - \boldsymbol{I}_j)$ is used to obtain

$$\widehat{\nabla}_{\text{2nd-order}} := \sum_{i=1}^{n} \sum_{j=1}^{n} \frac{\boldsymbol{\pi}_j}{2} (\frac{\partial f(\boldsymbol{I}_j)}{\partial \boldsymbol{I}_j} + \frac{\partial f(\boldsymbol{I}_i)}{\partial \boldsymbol{I}_i})(\boldsymbol{I}_i - \boldsymbol{I}_j) \frac{d\boldsymbol{\pi}_i}{d\boldsymbol{\theta}}. \tag{11}$$

Therefore, the bias of the Straight-Through estimator can be reduced if it can be modified to match the more accurate second-order approximation (Equation 11) in expectation. This is the motivation behind the ReinMax estimator given by

$$\widehat{\nabla}_{\text{ReinMax},\tau}(\boldsymbol{D}, \boldsymbol{\theta}) := 2\frac{\partial f(\boldsymbol{D})}{\partial \boldsymbol{D}}\left(\text{diag}\left(\frac{\boldsymbol{\pi}_\tau + \boldsymbol{D}}{2}\right) - \left(\frac{\boldsymbol{\pi}_\tau + \boldsymbol{D}}{2}\right)\left(\frac{\boldsymbol{\pi}_\tau + \boldsymbol{D}}{2}\right)^{\top}\right) - \frac{1}{2}\frac{\partial f(\boldsymbol{D})}{\partial \boldsymbol{D}}\left(\text{diag}(\boldsymbol{\pi}) - \boldsymbol{\pi}\boldsymbol{\pi}^{\top}\right). \tag{12}$$

When $\tau = 1$, the ReinMax estimator is equal to the second-order approximation in expectation:

$$\mathbb{E}_{\boldsymbol{D}\sim\boldsymbol{\pi}}[\widehat{\nabla}_{\text{ReinMax},\tau}(\boldsymbol{D}, \boldsymbol{\theta})] = \widehat{\nabla}_{\text{2nd-order}} \tag{13}$$

and so it has lower bias than the Straight-Through estimator. However, ReinMax has much higher variance than Straight-Through since the $\text{diag}\left(\frac{\boldsymbol{\pi}_\tau + \boldsymbol{D}}{2}\right) - \left(\frac{\boldsymbol{\pi}_\tau + \boldsymbol{D}}{2}\right)\left(\frac{\boldsymbol{\pi}_\tau + \boldsymbol{D}}{2}\right)^{\top}$ term depends on the random variable $\boldsymbol{D}$. We elaborate on this in Section 3.

## 2.5 Control Variates and REBAR (Tucker et al., 2017)

Control variates is a technique for reducing variance in Monte-Carlo estimators. Suppose we want to estimate $\mathbb{E}[h(\boldsymbol{x})]$ and $\mathbb{E}[g(\boldsymbol{x})]$ is known in closed-form. Then $\mathbb{E}[h(\boldsymbol{x})]$ can be estimated by

$$\widehat{h(\boldsymbol{x})} := h(\boldsymbol{x}) - \eta g(\boldsymbol{x}) + \eta E[g(\boldsymbol{x})], \tag{14}$$

where $g(\boldsymbol{x})$ is the control variate that needs to be chosen. The optimal value of $\eta = \frac{\text{Cov}(h(\boldsymbol{x}), g(\boldsymbol{x}))}{\text{Var}(g(\boldsymbol{x}))}$ yields $\text{Var}(\widehat{h(\boldsymbol{x})}) = (1 - \rho^2)\text{Var}(h(\boldsymbol{x}))$ where $\rho = \text{Corr}(h(\boldsymbol{x}), g(\boldsymbol{x}))$, implying that variance is reduced when the chosen control variate $g(\boldsymbol{x})$ correlates with $h(\boldsymbol{x})$. In practice, if a closed-form expression for $E[g(\boldsymbol{x})]$ is not known, it can be replaced with an unbiased low-variance estimator.

The REBAR estimator (Tucker et al., 2017) applies this technique to reduce the variance of the RE-INFORCE estimator (Williams, 1992), where $h(\boldsymbol{D}) := f(\boldsymbol{D})\nabla\log p(\boldsymbol{D})$. Their insight is that since $\boldsymbol{D}$ can be closely approximated by its Gumbel-softmax relaxation, simply replacing $\boldsymbol{D}$ with this relaxation yields a highly correlated control variate $g(\boldsymbol{G}) := f(\text{softmax}_\tau(\boldsymbol{\theta} + \boldsymbol{G}))\nabla\log p(\boldsymbol{G})$. Since $\mathbb{E}[g(\boldsymbol{G})]$ is not known in closed form, Tucker et al. (2017) show that it can be estimated with low variance via conditional marginalisation in the form of the Gumbel-Rao trick. The REBAR estimator can be written as

$$\widehat{\nabla}_{\text{REBAR},\tau}(\boldsymbol{\theta}, \boldsymbol{G}) = [f(\boldsymbol{D}) - \eta f(s_\tau(\boldsymbol{Y}_{\boldsymbol{\theta},\boldsymbol{D}}))]\nabla_{\boldsymbol{\theta}}\log p(\boldsymbol{D}) + \eta\nabla_{\boldsymbol{\theta}}f(s_\tau(\boldsymbol{\theta} + \boldsymbol{G})) - \eta\nabla_{\boldsymbol{\theta}}f(s_\tau(\boldsymbol{Y}_{\boldsymbol{\theta},\boldsymbol{D}})). \tag{15}$$

where $s_\tau(\cdot)$ is shorthand for $\text{softmax}_\tau(\cdot)$.

Table 1: Sample standard deviation of different estimators after 50 epochs of training a discrete VAE with latent size $8 \times 4$. The sample standard deviation is computed over 1024 samples of $\boldsymbol{D}$ while keeping the minibatch of input data fixed. We use $\tau = 1.3$ for all methods.

|  | ReinMax | Straight-Through | ReinMax-Argmax |
|---|---|---|---|
| Std | 7.400 | 2.733 | 2.873 |

## 3 Variance Reduction of ReinMax via Gumbel-Softmax Reparameterisation

We first pinpoint the source of high variance in ReinMax. By noticing that the $\operatorname{diag}(\cdot) - (\cdot)(\cdot)^\top$ terms in the ReinMax estimator (Equation 12) have the same form as the Jacobian of the $\operatorname{softmax}(\cdot)$ function, the ReinMax estimator can be rewritten in terms of two instances of the Straight-Through estimator evaluated at the original $\boldsymbol{\theta}$ and a new $\boldsymbol{\theta_D}$. That is,

$$\widehat{\nabla}_{\text{ReinMax},\tau}(\boldsymbol{D}, \boldsymbol{\theta}) = 2\widehat{\nabla}_{\text{ST},\tau}(\boldsymbol{D}, \boldsymbol{\theta_D}) - \frac{1}{2}\widehat{\nabla}_{\text{ST},\tau=1}(\boldsymbol{D}, \boldsymbol{\theta}) \tag{16}$$

where $\boldsymbol{\theta_D} = \log(\frac{\boldsymbol{\pi_\tau} + \boldsymbol{D}}{2})$ so that $\operatorname{softmax}(\boldsymbol{\theta_D}) = \frac{\boldsymbol{\pi_\tau} + \boldsymbol{D}}{2}$. Intuitively, the first term $\widehat{\nabla}_{\text{ST},\tau}(\boldsymbol{D}, \boldsymbol{\theta_D})$ suffers from high variance because $\boldsymbol{\theta_D}$ depends on the random variable $\boldsymbol{D}$ while in the second term $\boldsymbol{\theta}$ is constant.

To empirically verify that this is indeed the case, we modify ReinMax in Equation 16 by replacing the random variable $\boldsymbol{D}$ with the deterministic one-hot vector $\arg\max(\boldsymbol{\theta})$ in $\boldsymbol{\theta_D}$ and denote it by ReinMax-Argmax. Computing the variance for the ReinMax-Argmax, ReinMax and Straight-Through estimators on discrete VAEs (see Section 4 for experiment details) shows that ReinMax-Argmax reduces the variance of ReinMax to a level that is just slightly higher than Straight-Through, thus verifying our intuition (Table 1). Note that ReinMax-Argmax does not work in practice because replacing the random $\boldsymbol{D}$ term with the deterministic $\arg\max(\boldsymbol{\theta})$ significantly increases bias. Therefore, the goal of our methods is to reduce the variance of $\widehat{\nabla}_{\text{ST},\tau}(\boldsymbol{D}, \boldsymbol{\theta_D})$ without incurring a large bias.

### 3.1 ReinMax-Rao: Gumbel-Rao approximation of $\widehat{\nabla}_{\text{ST},\tau}(\boldsymbol{D}, \boldsymbol{\theta_D})$

Our ReinMax-Rao estimator is based on the intuition that the Straight-Through and Gumbel-Rao estimators are similar in the sense that they are approximations of the same quantity (i.e., the exact gradient), so it is reasonable to assume that replacing the high-variance first term of ReinMax $\widehat{\nabla}_{\text{ST},\tau}(\boldsymbol{D}, \boldsymbol{\theta_D})$ with the low-variance Gumbel-Rao estimator $\widehat{\nabla}_{\text{GR},\tau}(\boldsymbol{D}, \boldsymbol{\theta_D})$ also evaluated at $\boldsymbol{\theta_D}$ will not significantly alter the expectation of ReinMax. The ReinMax-Rao estimator is given by

$$\widehat{\nabla}_{\text{ReinMax-Rao},\tau}(\boldsymbol{D}, \boldsymbol{\theta}) = 2\widehat{\nabla}_{\text{GR},\tau}(\boldsymbol{D}, \boldsymbol{\theta_D}) - \frac{1}{2}\widehat{\nabla}_{\text{ST},\tau}(\boldsymbol{D}, \boldsymbol{\theta_D}) \tag{17}$$

and has lower variance but higher bias than ReinMax in practice.

### 3.2 ReinMax-CV: Bias Correction via Control Variates

Next, we take the idea of using $\widehat{\nabla}_{\text{GR},\tau}(\boldsymbol{D}, \boldsymbol{\theta_D})$ to reduce variance one step further by correcting the incurred bias of ReinMax-Rao using control variates. We again leverage the fact that the Straight-Through and Gumbel-Softmax estimators are approximations of the same quantity (i.e., the exact gradient) which suggests that they are strongly correlated and choose $\widehat{\nabla}_{\text{STGS},\tau}(\boldsymbol{G}, \boldsymbol{\theta_D})$ as the control variate for $\widehat{\nabla}_{\text{ST},\tau}(\boldsymbol{D}, \boldsymbol{\theta_D})$. We replace the $\widehat{\nabla}_{\text{ST},\tau}(\boldsymbol{D}, \boldsymbol{\theta_D})$ term in the ReinMax estimator with $\widehat{\nabla}_{\text{ST},\tau}(\boldsymbol{D}, \boldsymbol{\theta_D}) - \eta\widehat{\nabla}_{\text{STGS},\tau}(\boldsymbol{G}, \boldsymbol{\theta_D}) + \eta\mathbb{E}_{\boldsymbol{G}}[\widehat{\nabla}_{\text{STGS},\tau}(\boldsymbol{G}, \boldsymbol{\theta_D})]$. However, $\mathbb{E}_{\boldsymbol{G}}[\widehat{\nabla}_{\text{STGS},\tau}(\boldsymbol{G}, \boldsymbol{\theta_D})]$ is not known in closed form so we estimate it with the low-variance Gumbel-Rao estimator $\widehat{\nabla}_{\text{GR},\tau}(\boldsymbol{D}, \boldsymbol{\theta_D})$. In full, the ReinMax-CV estimator is given by

$$\widehat{\nabla}_{\text{ReinMax-CV},\tau}(\boldsymbol{G}, \boldsymbol{\theta}) = 2\widehat{\nabla}_{\text{ST},\tau=1}(\boldsymbol{D}, \boldsymbol{\theta_D}) - \eta\widehat{\nabla}_{\text{STGS},\tau}(\boldsymbol{G}, \boldsymbol{\theta_D}) + \eta\widehat{\nabla}_{\text{GR},\tau}(\boldsymbol{D}, \boldsymbol{\theta_D}) - \frac{1}{2}\widehat{\nabla}_{\text{ST},\tau=1}(\boldsymbol{D}, \boldsymbol{\theta}) \tag{18}$$

where $\tau$ and $\eta$ are hyperparameters. For simplicity, we only tune the temperature $\tau$ for the control variate terms and fix $\tau = 1$ for the first and last terms in Equation 18 which correspond to the original ReinMax terms. Note that in theory, the expectation of $\widehat{\nabla}_{\mathrm{STGS},\tau}(\boldsymbol{G}, \boldsymbol{\theta_D})$ and $\widehat{\nabla}_{\mathrm{GR},\tau}(\boldsymbol{D}, \boldsymbol{\theta_D})$ should be the same. However, our implementation of Gumbel-Rao follows existing ones where the derivative through the conditional reparameterisation is ignored. As a result, ReinMax-CV has higher bias than ReinMax in practice.

## 4 Experiments

In this section, we train VAEs (Kingma & Welling, 2014) with discrete latent spaces to evaluate the performance of the proposed ReinMax-Rao and ReinMax-CV estimators.

### 4.1 Variational Autoencoders

In a VAE, the exact log marginal likelihood $\log p_\psi(\boldsymbol{x})$ is intractable and cannot be used for learning. To sidestep this, a variational distribution $q_\phi(\boldsymbol{z}|\boldsymbol{x})$ is introduced to approximate the posterior $p_\psi(\boldsymbol{z}|\boldsymbol{x})$, resulting in the following training objective:

$$\log p(\boldsymbol{x}) = \mathbb{E}_{q_\phi(\boldsymbol{z}|\boldsymbol{x})}[\log p(\boldsymbol{x}, \boldsymbol{z}) - \log q_\phi(\boldsymbol{z}|\boldsymbol{x})] + D_{\mathrm{KL}}(q_\phi(\boldsymbol{z}|\boldsymbol{x})||p_\psi(\boldsymbol{z}|\boldsymbol{x})) \tag{19}$$

$$\geq \mathbb{E}_{q_\phi(\boldsymbol{z}|\boldsymbol{x})}[\log p(\boldsymbol{x}, \boldsymbol{z}) - \log q_\phi(\boldsymbol{z}|\boldsymbol{x})] \tag{20}$$

$$= \mathbb{E}_{q_\phi(\boldsymbol{z}|\boldsymbol{x})}[\log p_\psi(\boldsymbol{x}|\boldsymbol{z})] - D_{\mathrm{KL}}(q_\phi(\boldsymbol{z}|\boldsymbol{x})||p(\boldsymbol{z})) := L_{\mathrm{ELBO}}(\phi, \psi) \tag{21}$$

where $q_\phi(\boldsymbol{z}|\boldsymbol{x})$ and $p_\psi(\boldsymbol{x}|\boldsymbol{z})$ are neural networks parameterised by $\phi$ and $\psi$. The training objective is the Evidence Lower Bound (ELBO) given in Equation 21, and we focus on the discrete case where the prior $p(\boldsymbol{z})$ is set to a uniform categorical distribution.

### 4.2 Experiment set-up

We evaluate our estimators against Straight-Through (ST), Gapped Straight-Through (GST-1.0, Fan et al., 2022), ReinMax, Straight-Through Gumbel-Softmax (STGS) and Gumbel-Rao as baselines. Following previous work (Liu et al., 2023; Paulus et al., 2021; Jang et al., 2017), we train a VAE on discrete latent spaces of varying sizes on the MNIST dataset (LeCun et al., 1998) for 160 epochs with a batch size of 100. Refer to Appendix E for our hyperparameter settings. The VAE architecture consists of MLPs with ReLU activations, with hidden layer sizes of 512 and 256 for the encoder and hidden layers of sizes 256 and 512 for the decoder. Our implementation is based on the one provided by Liu et al. (2023) and is available at https://github.com/danielwang0452/DiscreteNumericalMethodsASC.

### 4.3 Results

We evaluate the sample bias and variance of the estimators over checkpoints (increments of 5 epochs from 0 to 50) of a discrete VAE trained with ReinMax on the smallest latent space setting (8 categorical dimensions and 4 latent dimensions). We focus on this small setting as it allows us to compute the exact gradient (Equation 2). For each gradient estimator, the bias is measured using the cosine similarity between the exact gradient and the sample mean of 1024 gradient estimates with a fixed batch of size 100 and fixed model parameters (Figure 1 [left]). The variance is simply the sample variance of 1024 gradient estimates (Figure 1 [right]).

In terms of bias, our ReinMax-Rao estimator has lower cosine similarity than ReinMax. This indicates that the Gumbel-Rao approximation used in ReinMax-Rao alters the expectation of ReinMax and increases its bias, but not to a significant extent as the cosine similarity is still higher than all other previous estimators. Because control variates preserve the expectation of an estimator while reducing its variance, in theory ReinMax-CV should have the same bias as ReinMax. However, the cosine similarity results reveal that this is not the case as the bias of ReinMax-CV sits between those of ReinMax and ReinMax-Rao. This is likely due to the implementation of Gumbel-Rao which ignores the derivative through the conditional reparameterisation, but despite it ReinMax-CV is still able to achieve lower bias than ReinMax-Rao.

In terms of variance, our ReinMax-Rao and ReinMax-CV estimators successfully reduce the variance of ReinMax with ReinMax-Rao having the lowest variance among the three ReinMax-style estimators. These results exhibit a clear bias-variance trade-off for the three ReinMax-style estimators with ReinMax having high variance and low bias, ReinMax-Rao having low variance but high bias and ReinMax-CV in the middle.

Tables 2 and 3 show that our ReinMax-Rao and ReinMax-CV estimators outperform previous estimators across most configurations. In particular, we observe that the configurations for which ReinMax-Rao achieves the best performance coincide with the configurations with the largest categorical dimension ($16 \times 12$ and $64 \times 8$). This indicates that low-variance estimators perform better on higher dimensions. On the other hand, it has been observed that the biased ReinMax estimator performs worse than unbiased estimators such as those based on REINFORCE on simpler problems with fewer dimensions (Liu et al., 2023). Together, these results suggest that low-bias high-variance estimators are more effective in simple low-dimensional settings, while high-bias low-variance estimators such as ours are more effective in complex high-dimensional settings.

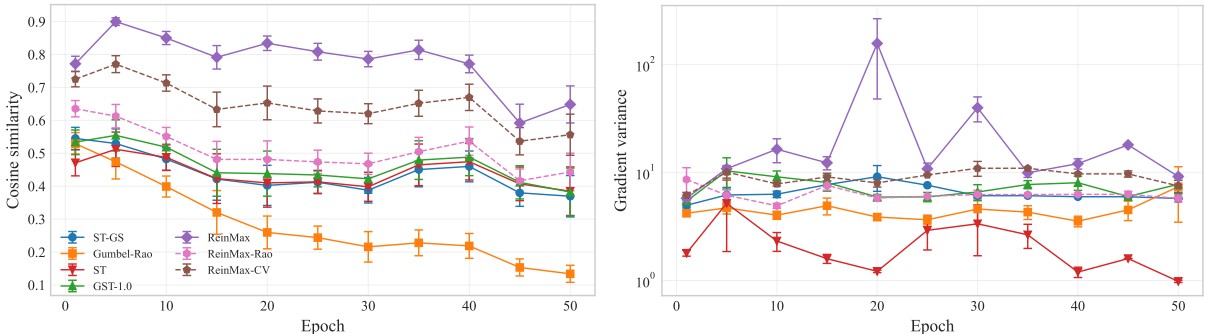

Figure 1: The sample bias and variance of the estimators over checkpoints of a discrete VAE trained with ReinMax, with 8 categorical dimensions and 4 latent dimensions. The bias is measured using the cosine similarity between the exact gradient and the sample mean of 1024 gradient estimates with a fixed batch of size 100 and fixed model parameters. The variance is simply the sample variance of 1024 gradient estimates. All the temperatures here are 1, except $\tau = 0.1$ for ReinMax-CV.

Table 2: Train negative ELBO ($\downarrow$) across different configurations of categorical dimension $\times$ latent dimension. Best results are in **bold**, second best are underlined, third best are *italic*.

| Method | $8\times4$ | $4\times24$ | $8\times16$ | $16\times12$ | $64\times8$ | $10\times30$ |
|---|---|---|---|---|---|---|
| Gumbel | $127.59\pm0.23$ | $101.49\pm0.09$ | $99.20\pm0.14$ | $100.42\pm0.15$ | $103.87\pm0.13$ | $99.80\pm0.11$ |
| Gumbel-Rao | $125.65\pm0.14$ | $100.14\pm0.12$ | $99.63\pm0.23$ | $109.72\pm6.83$ | $112.48\pm0.18$ | $102.28\pm0.19$ |
| ST | $136.40\pm0.41$ | $112.21\pm0.19$ | $113.32\pm0.09$ | $113.31\pm0.08$ | $113.97\pm0.07$ | $136.73\pm0.40$ |
| GST-1.0 | $128.31\pm0.21$ | $101.64\pm0.11$ | $98.31\pm0.16$ | $98.37\pm0.17$ | $102.41\pm0.16$ | $98.66\pm0.14$ |
| ReinMax | $125.08\pm0.20$ | **$99.88\pm0.08$** | $97.80\pm0.09$ | $97.98\pm0.13$ | *$101.09\pm0.11$* | $98.50\pm0.13$ |
| ReinMax-Rao | *$125.31\pm0.28$* | $100.41\pm0.12$ | *$98.03\pm0.14$* | **$97.62\pm0.09$** | **$100.24\pm0.16$** | *$98.61\pm0.16$* |
| ReinMax-CV | **$124.94\pm0.24$** | *$100.19\pm0.13$* | **$97.72\pm0.07$** | *$98.21\pm0.12$* | $100.66\pm0.11$ | **$98.07\pm0.15$** |

## 5 Understanding ReinMax

The ReinMax estimator is presented from a numerical ODE perspective in Liu et al. (2023), where it is derived via an approximation using Heun's method which is an instance of a second-order Runge-Kutta method for solving ODEs. Therefore, it might be possible to derive better gradient estimators by leveraging alternative numerical ODE methods. We investigate this by generalising the construction of ReinMax to the whole family of second-order Runge-Kutta methods based on the fact that in the numerical ODE setting, there is no objective preference of one method over another which may allow us to select alternative second-order Runge-Kutta approximations that outperform Heun's method. However, our results show that Heun's

Table 3: Test negative ELBO (↓) across different configurations of categorical dimension × latent dimension. Best results are in **bold**, second best are underlined, third best are *italic*.

| Method | 8×4 | 4×24 | 8×16 | 16×12 | 64×8 | 10×30 |
|---|---|---|---|---|---|---|
| Gumbel | 128.72±0.22 | 103.80±0.11 | 101.37±0.16 | 102.71±0.13 | 106.03±0.15 | 101.74±0.11 |
| Gumbel-Rao | 127.62±0.12 | *102.88±0.15* | 102.07±0.23 | 111.28±6.70 | 113.71±0.17 | 103.97±0.21 |
| ST | 137.01±0.45 | 113.63±0.20 | 114.81±0.10 | 114.46±0.09 | 115.45±0.10 | 136.45±0.61 |
| GST-1.0 | 129.15±0.15 | 104.04±0.11 | 101.27±0.14 | 101.67±0.20 | 105.34±0.18 | 100.77±0.12 |
| ReinMax | *127.27±0.16* | **102.26±0.06** | 100.73±0.10 | *100.86±0.13* | *103.35±0.10* | *100.80±0.13* |
| ReinMax-Rao | **126.60±0.28** | 103.31±0.13 | *100.74±0.14* | **100.32±0.11** | **102.76±0.19** | 101.01±0.18 |
| ReinMax-CV | 126.63±0.23 | 102.45±0.13 | **100.16±0.07** | 100.57±0.13 | 102.96±0.14 | **100.49±0.15** |

method outperforms all other second-order Runge-Kutta methods in practice. We explain this by arguing that a simpler but mathematically equivalent toolkit of numerical integration methods is more suitable for this problem than numerical ODE methods. From this perspective, we conclude by discussing the difficulties of constructing more accurate gradient approximations.

## 5.1 Straight-Through and ReinMax: A Numerical ODE Perspective

We first provide the background of Straight-Through and ReinMax from a numerical ODE perspective, as presented by Liu et al. (2023). Consider an ODE of the form $\frac{dy}{dt} = \lambda(t, y)$ with the initial condition $y(0) = y_0$. The Forward Euler method iteratively approximates the solution via the equation

$$y_{n+1} = y_n + h\lambda(t_n, y_n) \tag{22}$$

which can be interpreted as estimating the next point $y_{n+1}$ by taking the slope at $(t_n, y_n)$ and taking a step of chosen size $h$ along the slope. The second-order Runge-Kutta methods are given by

$$y_{n+1} = y_n + h\left((1 - \frac{1}{2\alpha})\lambda(t_n, y_n) + \frac{1}{2\alpha}\lambda(t_n + \alpha h, y_n + \alpha h\lambda(t_n, y_n))\right) \tag{23}$$

and can be interpreted as refining the Forward Euler method by estimating the slope at the next point $(t_n + \alpha h, y_n + \alpha h\lambda(t_n, y_n))$, and weighting it with the original slope at $(t_n, y_n)$ through the parameter $\alpha$ before taking a step along the slope to obtain $y_{n+1}$. For any value of $\alpha$, the Taylor series of the Runge-Kutta approximation matches the Taylor series of the true function up to the second-order term, and as such, there is no canonical choice for $\alpha$. In practice, the most commonly used instances of second-order Runge-Kutta methods are Heun's method, Ralston's method, and the midpoint method corresponding to $\alpha = 1$, $\frac{2}{3}$, and $\frac{1}{2}$ respectively.

In order to derive the first-order and second-order approximations in Equation 9 and Equation 11, we define a function $g : [0, 1] \to \mathbb{R}$ by $g(x) := f(x \cdot \boldsymbol{I}_i + (1 - x) \cdot \boldsymbol{I}_j)$ so that $g(0) = f(\boldsymbol{I}_i)$ and $g(1) = f(\boldsymbol{I}_j)$ and the derivatives are given by $g'(0) = \frac{\partial f(\boldsymbol{I}_i)}{\partial \boldsymbol{I}_i}$ and $g'(1) = \frac{\partial f(\boldsymbol{I}_j)}{\partial \boldsymbol{I}_j}$. By setting $\lambda(t, y) = g'(t)$ with $g(0) = f(\boldsymbol{I}_j)$, a step size of $h = 1$ and $n = 1$ iteration in Equation 22 and $\alpha = 1$ in Equation 23, we have the first-order approximation $g(1) - g(0) \approx g'(0)(1 - 0)$ which is equivalent to $f(\boldsymbol{I}_i) - f(\boldsymbol{I}_j) \approx \frac{\partial f(\boldsymbol{I}_j)}{\partial \boldsymbol{I}_j}(\boldsymbol{I}_i - \boldsymbol{I}_j)$ and the second-order approximation $g(1) - g(0) \approx \frac{1}{2}(g'(0) + g'(1))(1 - 0)$ which is equivalent to $f(\boldsymbol{I}_i) - f(\boldsymbol{I}_j) \approx \frac{1}{2}(\frac{\partial f(\boldsymbol{I}_i)}{\partial \boldsymbol{I}_i} + \frac{\partial f(\boldsymbol{I}_j)}{\partial \boldsymbol{I}_j})(\boldsymbol{I}_i - \boldsymbol{I}_j)$ by the definition of $g$.

## 5.2 Generalisation of ReinMax to 2nd-Order Runge-Kutta Methods

The above construction can be extended to the family of second-order Runge-Kutta methods by expressing it in terms of the parameter $\beta = 1 - \frac{1}{2\alpha}$:

$$g(1) - g(0) \approx (1 - 0)(\beta g'(0) + (1 - \beta)g'(1)). \tag{24}$$

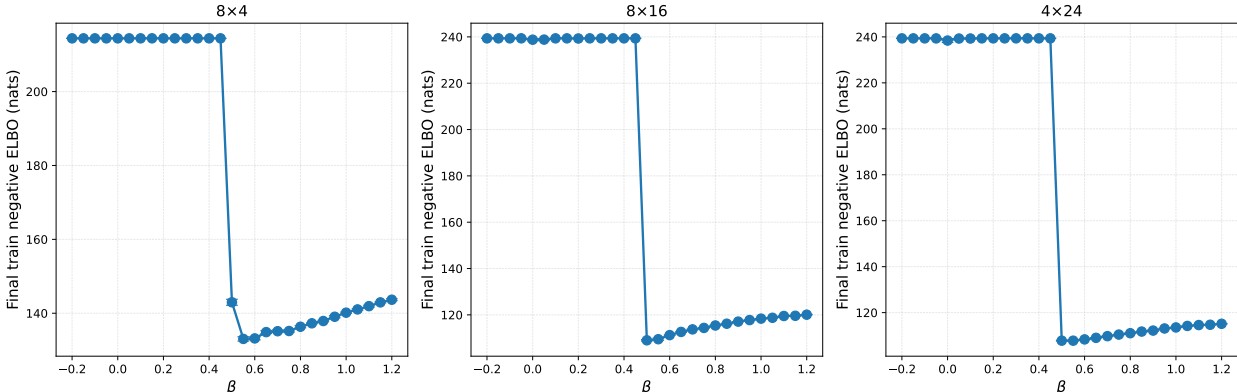

Figure 2: The negative ELBO on the train set for the $8 \times 4$, $8 \times 16$ and $4 \times 24$ VAEs after 50 epochs trained with $\widehat{\nabla}_{\text{ReinMax-RK2},\beta}$ as a function of $\beta$, spaced evenly from -0.2 to 1.2 in increments of 0.05. The minimum is achieved at approximately $\beta = 0.5$ which corresponds to the original ReinMax estimator. We use $\tau = 1$ here.

Again by substituting in the definition of $g$, this equivalent to

$$f(\boldsymbol{I}_i) - f(\boldsymbol{I}_j) \approx ((1 - \beta)\frac{\partial f(\boldsymbol{I}_j)}{\partial \boldsymbol{I}_j} + \beta\frac{\partial f(\boldsymbol{I}_i)}{\partial \boldsymbol{I}_i})(\boldsymbol{I}_i - \boldsymbol{I}_j). \tag{25}$$

By choosing $\beta \in [0,1]$, the $\beta$ parameter can be interpreted as reweighting the endpoints with Heun's method recovered by $\beta = \frac{1}{2}$. Using this $\beta$ parameterisation, we define the general second-order Runge-Kutta approximation analogously to Equation 11 by

$$\widehat{\nabla}_{\text{RK2},\beta} := \sum_{i=1}^{n}\sum_{j=1}^{n} \boldsymbol{\pi}_j((1 - \beta)\frac{\partial f(\boldsymbol{I}_j)}{\partial \boldsymbol{I}_j} + \beta\frac{\partial f(\boldsymbol{I}_i)}{\partial \boldsymbol{I}_i})(\boldsymbol{I}_i - \boldsymbol{I}_j)\frac{d\,\boldsymbol{\pi}_i}{d\,\boldsymbol{\theta}}. \tag{26}$$

We now define the ReinMax-RK2 estimator by

$$\widehat{\nabla}_{\text{ReinMax-RK2},\beta}(\boldsymbol{D},\boldsymbol{\theta}) := 2\frac{\partial f(\boldsymbol{D})}{\partial \boldsymbol{D}}\left(\text{diag}\left(\frac{\boldsymbol{\pi}+\boldsymbol{D}}{2}\right) - \left((\beta\boldsymbol{\pi} + (1-\beta)\boldsymbol{D})\right)\left(\frac{\boldsymbol{\pi}+\boldsymbol{D}}{2}\right)^{\top}\right) - \beta\widehat{\nabla}_{\text{ST},\tau=1}(\boldsymbol{D},\boldsymbol{\theta}) \tag{27}$$

and show that $\widehat{\nabla}_{\text{ReinMax-RK2},\beta}(\boldsymbol{D},\boldsymbol{\theta})$ is equal to $\widehat{\nabla}_{\text{RK2},\beta}$ in expectation by extending the proof given by Liu et al. (2023) to incorporate the $\beta$ parameter.

**Theorem 1.**

$$\mathbb{E}[\widehat{\nabla}_{\textit{ReinMax-RK2},\beta}] = \widehat{\nabla}_{\textit{RK2},\beta}.$$

The proof is given in Appendix A. We investigate how the value of $\beta$ affects the performance of the $\widehat{\nabla}_{\text{ReinMax}-RK2,\beta}$ on training discrete VAEs with the results shown in Figure 2. Given that the family of second-order Runge-Kutta methods does not have a preference for the value of $\beta$, it is unintuitive that the best performance is attained at $\beta = \frac{1}{2}$ which coincides with the original ReinMax estimator. We now present an alternative perspective of the Straight-Through and ReinMax estimators from a simpler numerical integration perspective which provides an explanation for why ReinMax achieves the best performance at $\beta = \frac{1}{2}$.

### 5.3 Straight-Through and ReinMax: a Numerical Integration Perspective

The construction in Section 5.1 suggests that it may be possible to derive better gradient estimators by considering more sophisticated numerical ODE methods than Heun's method. However, we argue that

numerical ODE methods are not well suited to this problem because they deal with non-autonomous ODEs where the function $\lambda(t, y)$ depends on two arguments, but we have set $\lambda(t, y) = g'(t)$ to only depend on one argument. Hence, the core mechanism of second-order Runge-Kutta methods which is to estimate the slope at the next point $(t_n + \alpha h, y_n + \alpha h \lambda(t_n, y_n))$ is redundant. Furthermore, Runge-Kutta methods are designed to generate a sequence of points over small intervals to approximate an unknown function, whereas we are only interested in approximating the numerical value $g(1) - g(0)$.

These problems can be avoided by viewing the approximation of $g(1) - g(0)$ from a numerical integration perspective, where the task is to approximate the integral $\int_0^1 g'(x)dx$ given the endpoints $g'(0)$ and $g'(1)$. Geometrically, the first-order approximation $g(1) - g(0) \approx g'(0)(1 - 0)$ estimates $g'(x)$ from 0 to 1 by a horizontal line at height $g'(0)$ and the integral $\int_0^1 g'(x)dx$ is simply approximated by a rectangle of height $g'(0)$ and width 1. The second-order approximation $g(1) - g(0) \approx \frac{1}{2}(g'(0) + g'(1))(1 - 0)$ interpolates $g'(x)$ by a straight line from $g'(0)$ to $g'(1)$ which means the integral $\int_0^1 g'(x)dx$ is approximated by the trapezoidal rule. From this perspective, we can understand why ReinMax-RK2 attains the best performance at $\beta = \frac{1}{2}$ in Figure 2. The general $\beta$-parameterised approximation $g(1) - g(0) \approx (1 - 0)(\beta g'(0) + (1 - \beta)g'(1))$ can be visualised geometrically as the area under the straight line with endpoints $[0, 2\beta g'(0)]$ and $[1, 2(1 - \beta)g'(1)]$, with $\beta = \frac{1}{2}$ recovering the trapezoidal rule where the endpoints of the line are exactly $[0, g'(0)]$ and $[1, g'(1)]$. However, for other values of $\beta$, the endpoints of the interval are shifted up and down respectively so that they no longer lie on the endpoints of the curve $g'(x)$ that we wish to approximate. Therefore, the approximation becomes more inaccurate as $\beta$ moves away from $\frac{1}{2}$.

Although the mathematical expressions are identical, we have simplified the conceptual interpretation of the second-order approximation from Heun's method to the trapezoidal rule. We now consider whether more sophisticated numerical integration methods can be used to derive more accurate gradient approximations. The first-order and second-order approximations of $g'(x)$ use degree 0 (rectangle rule) and 1 (trapezoidal rule) polynomials respectively, and a natural extension of this is to consider higher-degree polynomials. A quadratic approximation of $g'(x)$ is the well-known Simpson's Rule which interpolates $g'(x)$ by the unique parabola passing through its endpoints $g'(0)$, $g'(1)$ and midpoint $g'(\frac{1}{2})$, from which the approximate integral has an analytic expression. However, this method involves the term $g'(\frac{1}{2}) = \frac{\partial f(\boldsymbol{I}_{ij})}{\partial \boldsymbol{I}_{ij}}$ where $\boldsymbol{I}_{ij} := \frac{\boldsymbol{I}_i + \boldsymbol{I}_j}{2}$, which is problematic in two ways. Firstly, in principle $f$ should only be evaluated with categorical one-hot vectors as inputs which is violated by $\boldsymbol{I}_{ij}$ when $i \neq j$. Secondly, constructing an estimator that in expectation involves summation over the $\frac{\partial f(\boldsymbol{I}_{ij})}{\partial \boldsymbol{I}_{ij}}$ term in addition to the existing $\frac{\partial f(\boldsymbol{I}_i)}{\partial \boldsymbol{I}_i}$ and $\frac{\partial f(\boldsymbol{I}_j)}{\partial \boldsymbol{I}_j}$ terms is complicated. Alternatively, we can consider a degree 3 polynomial given by the cubic spline interpolation method. The idea behind this method is to interpolate between $g'(0)$ and $g'(1)$ by the unique cubic polynomial that has derivative $g''(0)$ and $g''(1)$ at the endpoints 0 and 1 respectively, from which the area under the cubic spline can be derived analytically. While this method does not require $g'(x)$ evaluated at any point between 0 and 1 unlike Simpson's Rule, it requires computing the Hessian $\frac{\partial^2 f(\boldsymbol{D})}{\partial \boldsymbol{D}^2}$ which is impractical in modern deep learning applications. Thus, we conclude that given only $g'(0)$ and $g'(1)$, the trapezoidal rule which naturally draws a straight line between the two points is in some sense the best computationally viable approximation that can be made without any additional information.

## 6    Conclusion

We introduced the ReinMax-Rao and ReinMax-CV estimators which extend the ReinMax estimator using control variates and Rao-Blackwellisation techniques. Despite incurring larger biases than ReinMax, our estimators have lower variance which results in better performance on training discrete VAEs. We explored a possible approach to reduce the bias of ReinMax using alternative numerical ODE methods to derive the gradient approximation, specifically by generalising it to the family of second-order Runge-Kutta methods. However, this did not work in practice which can be explained by viewing the approximation from a simpler numerical integration perspective. From this perspective, we argue that constructing more accurate approximations requires a different toolkit of numerical methods and doing so in a computationally efficient manner remains a challenging problem.

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

## A Proof of Theorem 1.

**Theorem 1**

$$\mathbb{E}[\widehat{\nabla}_{ReinMax\text{-}RK2,\beta}] = \widehat{\nabla}_{RK2,\beta}.$$

**Proof** We will first derive the $k^{\text{th}}$ entry of $\widehat{\nabla}_{\text{ReinMax-RK2},\beta}$, and show that in expectation it is equal to the $k^{\text{th}}$ entry of $\widehat{\nabla}_{\text{RK2},\beta}$.

In both terms of $\widehat{\nabla}_{\text{ReinMax-RK2},\beta}$ (Equation 27), the $k^{\text{th}}$ entry is given by a dot product between $\frac{\partial f(\boldsymbol{D})}{\partial \boldsymbol{D}}$ and the $k^{\text{th}}$ column of the term of the form $\text{diag}(\boldsymbol{\pi}) - \boldsymbol{\pi}\boldsymbol{\pi}^{\mathsf{T}}$. For the second term, the $k^{\text{th}}$ column of the matrix $\text{diag}(\boldsymbol{\pi}) - \boldsymbol{\pi}\boldsymbol{\pi}^{\mathsf{T}}$ is $\boldsymbol{\pi}_k(\boldsymbol{I}_k - \boldsymbol{\pi})$. For the first term, the $k^{\text{th}}$ column of the matrix $\text{diag}(\boldsymbol{\pi}_{\boldsymbol{D}}) - \boldsymbol{\pi}_\beta \boldsymbol{\pi}_{\boldsymbol{D}}^{\mathsf{T}}$ is $(\boldsymbol{\pi}_{\boldsymbol{D}})_k \boldsymbol{I}_k - (\boldsymbol{\pi}_{\boldsymbol{D}})_k \boldsymbol{\pi}_\beta = \frac{\boldsymbol{\pi}_k + \delta_{\boldsymbol{DI}_k}}{2}(\boldsymbol{I}_k - (\beta\boldsymbol{\pi} + (1-\beta)\boldsymbol{D}))$ where $\boldsymbol{\pi}_{\boldsymbol{D}} = \frac{\boldsymbol{\pi} + \boldsymbol{D}}{2}$, $\boldsymbol{\pi}_\beta = \beta\boldsymbol{\pi} + (1-\beta)\boldsymbol{D}$ and $\delta_{\boldsymbol{DI}_k}$ is the indicator function of the event $\boldsymbol{D} = \boldsymbol{I}_k$. Putting these together, we have

$$\mathbb{E}[\widehat{\nabla}_{\text{ReinMax-RK2},\beta,k}] = \mathbb{E}_{\boldsymbol{D}\sim\boldsymbol{\pi}}\left[\frac{\partial f(\boldsymbol{D})}{\partial \boldsymbol{D}}(2 \cdot \frac{\boldsymbol{\pi}_k + \delta_{\boldsymbol{DI}_k}}{2}(\boldsymbol{I}_k - (\beta\boldsymbol{\pi} + (1-\beta)\boldsymbol{D})) - \beta\boldsymbol{\pi}_k(\boldsymbol{I}_k - \boldsymbol{\pi}))\right] \quad (28)$$

$$= \mathbb{E}_{\boldsymbol{D}\sim\boldsymbol{\pi}}\left[\frac{\partial f(\boldsymbol{D})}{\partial \boldsymbol{D}}(\boldsymbol{\pi}_k \boldsymbol{I}_k - (1-\beta)\boldsymbol{\pi}_k \boldsymbol{D} + \delta_{\boldsymbol{DI}_k}(\boldsymbol{I}_k - \beta\boldsymbol{\pi} - (1-\beta)\boldsymbol{D}) - \beta\boldsymbol{\pi}_k \boldsymbol{I}_k)\right] \quad (29)$$

$$= \mathbb{E}_{\boldsymbol{D}\sim\boldsymbol{\pi}}\left[\frac{\partial f(\boldsymbol{D})}{\partial \boldsymbol{D}}((1-\beta)\boldsymbol{\pi}_k \boldsymbol{I}_k - (1-\beta)\boldsymbol{\pi}_k \boldsymbol{D} + \delta_{\boldsymbol{DI}_k}(\beta\boldsymbol{I}_k - \beta\boldsymbol{\pi}))\right] \quad (30)$$

$$= \mathbb{E}_{\boldsymbol{D}\sim\boldsymbol{\pi}}\left[\frac{\partial f(\boldsymbol{D})}{\partial \boldsymbol{D}}((1-\beta)\boldsymbol{\pi}_k(\boldsymbol{I}_k - \boldsymbol{D}) + \beta\delta_{\boldsymbol{DI}_k}(\boldsymbol{I}_k - \boldsymbol{\pi}))\right] \quad (31)$$

$$= (1-\beta)\mathbb{E}_{\boldsymbol{D}\sim\boldsymbol{\pi}}\left[\boldsymbol{\pi}_k\frac{\partial f(\boldsymbol{D})}{\partial \boldsymbol{D}}(\boldsymbol{I}_k - \boldsymbol{D})\right] + \beta\mathbb{E}_{\boldsymbol{D}\sim\boldsymbol{\pi}}\left[\frac{\partial f(\boldsymbol{D})}{\partial \boldsymbol{D}}\delta_{\boldsymbol{DI}_k}(\boldsymbol{D} - \boldsymbol{\pi})\right]. \quad (32)$$

Now we proceed from the other direction:

$$\widehat{\nabla}_{\text{RK2},\beta,k} = \sum_i \sum_j \boldsymbol{\pi}_j((1-\beta)\frac{\partial f(\boldsymbol{I}_j)}{\partial \boldsymbol{I}_j} + \beta\frac{\partial f(\boldsymbol{I}_i)}{\partial \boldsymbol{I}_i})(\boldsymbol{I}_i - \boldsymbol{I}_j)\frac{d\boldsymbol{\pi}_i}{d\boldsymbol{\theta}_k} \quad (33)$$

$$= \sum_i \sum_j \boldsymbol{\pi}_j\boldsymbol{\pi}_i(\delta_{ik} - \boldsymbol{\pi}_k)((1-\beta)\frac{\partial f(\boldsymbol{I}_j)}{\partial \boldsymbol{I}_j} + \beta\frac{\partial f(\boldsymbol{I}_i)}{\partial \boldsymbol{I}_i})(\boldsymbol{I}_i - \boldsymbol{I}_j) \quad (34)$$

$$= \sum_j \boldsymbol{\pi}_j\boldsymbol{\pi}_k((1-\beta)\frac{\partial f(\boldsymbol{I}_j)}{\partial \boldsymbol{I}_j} + \beta\frac{\partial f(\boldsymbol{I}_k)}{\partial \boldsymbol{I}_k})(\boldsymbol{I}_k - \boldsymbol{I}_j) \quad (35)$$

$$= (1-\beta)\sum_j \boldsymbol{\pi}_j\boldsymbol{\pi}_k\frac{\partial f(\boldsymbol{I}_j)}{\partial \boldsymbol{I}_j}(\boldsymbol{I}_k - \boldsymbol{I}_j) + \beta\boldsymbol{\pi}_k\frac{\partial f(\boldsymbol{I}_k)}{\partial \boldsymbol{I}_k}(\boldsymbol{I}_k - \sum_j \boldsymbol{\pi}_j\boldsymbol{I}_j) \quad (36)$$

$$= (1-\beta)\mathbb{E}_{\boldsymbol{D}\sim\boldsymbol{\pi}}\left[\boldsymbol{\pi}_k\frac{\partial f(\boldsymbol{D})}{\partial \boldsymbol{D}}(\boldsymbol{I}_k - \boldsymbol{D})\right] + \beta\mathbb{E}_{\boldsymbol{D}\sim\boldsymbol{\pi}}\left[\frac{\partial f(\boldsymbol{D})}{\partial \boldsymbol{D}}\delta_{\boldsymbol{DI}_k}(\boldsymbol{D} - \boldsymbol{\pi})\right]. \quad (37)$$

Putting both sides together, we have

$$\mathbb{E}[\widehat{\nabla}_{\text{ReinMax-RK2},\beta,k}] = \widehat{\nabla}_{\text{RK2},\beta,k}$$

## B Implementation Pseudocode

Here we denote the tempered softmax Jacobian by $J_\tau(\boldsymbol{\theta}) := \frac{d\text{softmax}_\tau(\boldsymbol{\theta})}{d\boldsymbol{\theta}} = (\frac{1}{t}(\text{diag}(\text{softmax}_\tau(\boldsymbol{\theta})) - \text{softmax}_\tau(\boldsymbol{\theta})\text{softmax}_\tau(\boldsymbol{\theta})^{\mathsf{T}})$.

---

**Algorithm 1** ReinMax-Rao (equation 17)

---

1: **Input:** Temperature $\tau$
2: ▷ *Forward Pass*
3: Sample $\boldsymbol{D} \sim \text{softmax}(\boldsymbol{\theta})$
4: $\mathcal{L} \leftarrow f(\boldsymbol{D})$                                                      ▷ compute scalar loss
5: ▷ *Backward Pass*
6: Compute $\frac{\partial f(\boldsymbol{D})}{\partial \boldsymbol{D}}$ via backpropagation
7: $\boldsymbol{\theta_D} \leftarrow \log\left(\frac{\boldsymbol{\pi}+\boldsymbol{D}}{2}\right)$
8: Sample $(\boldsymbol{Y}_{\boldsymbol{\theta_D},\boldsymbol{D},k})_{k=1}^{K}$                    ▷ sampled according to equation 7
9: $J_1 \leftarrow \frac{1}{K}\sum_{k=1}^{K} J_\tau(\boldsymbol{Y}_{\boldsymbol{\theta_D},\boldsymbol{D},k})$          ▷ corresponding to $\widehat{\nabla}_{\text{GR},\tau}(\boldsymbol{D},\boldsymbol{\theta_D})$
10: $J_2 \leftarrow J_\tau(\boldsymbol{\theta})$                                        ▷ corresponding to $\widehat{\nabla}_{\text{ST},\tau}(\boldsymbol{D},\boldsymbol{\theta})$
11: $\frac{\partial f(\boldsymbol{D})}{\partial \boldsymbol{\theta}} \leftarrow \frac{\partial f(\boldsymbol{D})}{\partial \boldsymbol{D}}\left(2J_1 - \frac{1}{2}J_2\right)$
12: **Output:** $\frac{\partial f(\boldsymbol{D})}{\partial \boldsymbol{\theta}}$

---

---

**Algorithm 2** ReinMax-CV (equation 18)

---

1: **Input:** Temperature $\tau$, $\eta$
2: ▷ *Forward Pass*
3: Sample $\boldsymbol{G} \sim \text{Gumbel}(0,1)$
4: $\boldsymbol{D} \leftarrow$ one-hot $\arg\max(\boldsymbol{\theta}+\boldsymbol{G})$
5: $\mathcal{L} \leftarrow f(\boldsymbol{D})$                                                      ▷ compute scalar loss
6: ▷ *Backward Pass*
7: Compute $\frac{\partial f(\boldsymbol{D})}{\partial \boldsymbol{D}}$ via backpropagation
8: $\boldsymbol{\theta_D} \leftarrow \log\left(\frac{\boldsymbol{\pi}+\boldsymbol{D}}{2}\right)$
9: $J_1 \leftarrow J_{\tau=1}(\boldsymbol{\theta_D})$                                    ▷ corresponding to $\widehat{\nabla}_{\text{ST},\tau=1}(\boldsymbol{D},\boldsymbol{\theta_D})$
10: $J_2 \leftarrow J_\tau(\boldsymbol{\theta_D}+\boldsymbol{G})$                              ▷ corresponding to $\widehat{\nabla}_{\text{STGS},\tau}(\boldsymbol{G},\boldsymbol{\theta_D})$
11: Sample $(\boldsymbol{Y}_{\boldsymbol{\theta_D},\boldsymbol{D},k})_{k=1}^{K}$                    ▷ sampled according to equation 7
12: $J_3 \leftarrow \frac{1}{K}\sum_{k=1}^{K} J_\tau(\boldsymbol{Y}_{\boldsymbol{\theta_D},\boldsymbol{D},k})$          ▷ corresponding to $\widehat{\nabla}_{\text{GR},\tau}(\boldsymbol{D},\boldsymbol{\theta_D})$
13: $J_4 \leftarrow J_{\tau=1}(\boldsymbol{\theta})$                                      ▷ corresponding to $\widehat{\nabla}_{\text{ST},\tau=1}(\boldsymbol{D},\boldsymbol{\theta})$
14: $\frac{\partial f(\boldsymbol{D})}{\partial \boldsymbol{\theta}} \leftarrow \frac{\partial f(\boldsymbol{D})}{\partial \boldsymbol{D}}\left(2J_1 - \eta J_2 + \eta J_3 - \frac{1}{2}J_4\right)$
15: **Output:** $\frac{\partial f(\boldsymbol{D})}{\partial \boldsymbol{\theta}}$

---

## C   Metrics

Here we provide the precise formulation of the Cosine Similarity and Gradient Variance metrics shown in Figure 1. Let $\hat{g}_i$ be the gradient estimate for the $i^{th}$ sample of $\boldsymbol{D}$, and $\nabla L$ be the exact gradient computed according to equation 2. The cosine similarity is given by

$$\text{Cosine Similarity} := \frac{\overline{g}^{\mathsf{T}}\nabla L}{\|\overline{g}\|\|\nabla L\|}. \tag{38}$$

where $\overline{g} := \frac{1}{N}\sum_{i=1}^{N}\hat{g}_i$ is the sample mean over $N = 1000$ samples of $\boldsymbol{D}$.

The variance is obtained by first computing the sample variance over samples of $\boldsymbol{D}$:

$$g_{var} := \frac{1}{N}\sum_{i=1}^{N}\hat{g}_i^{\circ 2} - \overline{g}^{\circ 2} \tag{39}$$

where $\circ 2$ denotes the element-wise square. We normalise this by $\overline{g}_b$ to get

$$\text{Gradient Variance} := \frac{\|g_{var}\|}{\|\overline{g}\|}. \tag{40}$$

Finally, these quantities are averaged over a fixed batch of 100 data points from the training dataset.

## D  Computational Cost

Table 4: Average time per epoch for each estimator on the 8×4 VAE with settings described in section 4.2 .

|  | Gumbel | ReinMax | ST | GST-1.0 | Gumbel-Rao | ReinMax-Rao | ReinMax-CV |
|---|---|---|---|---|---|---|---|
| Time per epoch | 3.45s | 3.62s | 3.79s | 4.00s | 6.51s | 7.55s | 7.57s |

## E  Hyperparameters

For ReinMax-Rao, we follow Liu et al. (2023) by conducting a full grid search with the Adam Kingma & Ba (2015) and RAdam Liu et al. (2020) optimisers, learning rates {0.001, 0.0007, 0.0005, 0.0003}, and temperatures {0.1, 0.3, 0.5, 0.7, 1.0, 1.1, 1.2, 1.3, 1.4, 1.5}. For ReinMax-CV, we select the same optimiser and learning rate used by ReinMax for the corresponding VAE latent space configuration and conduct a full grid search on the temperatures {0.1, 0.3, 0.5, 0.7, 1.0, 1.1, 1.2, 1.3, 1.4, 1.5} and $\eta \in$ {0.1, 0.3, 0.5, 0.7, 1.0, 1.5, 2.0}. For all other estimators, we use the settings given by Liu et al. (2023) as the experiment set-up is the same. For the estimators that utilise the Gumbel-Rao Monte-Carlo approximation (Equation 6), we set $K = 100$. We use 10 seeds for each experiment. The hyperparameters used for each estimator and VAE setting are shown in Table 5.

Table 5: Hyperparameters used for the VAEs of varying latent space sizes.

|  |  | $8 \times 4$ | $4 \times 24$ | $8 \times 16$ | $16 \times 12$ | $64 \times 8$ | $10 \times 30$ |
|---|---|---|---|---|---|---|---|
| STGS | Optimizer | Adam | RAdam | RAdam | RAdam | RAdam | RAdam |
|  | Learning Rate | 0.0003 | 0.0005 | 0.0005 | 0.0007 | 0.0007 | 0.0005 |
|  | Temperature | 0.5 | 0.3 | 0.5 | 0.7 | 0.7 | 0.5 |
| GR | Optimizer | Adam | RAdam | RAdam | Adam | Adam | RAdam |
|  | Learning Rate | 0.0005 | 0.0005 | 0.0007 | 0.0005 | 0.0007 | 0.0005 |
|  | Temperature | 0.5 | 0.3 | 0.7 | 1.0 | 2.0 | 1.0 |
| ST | Optimizer | Adam | RAdam | RAdam | Adam | Adam | RAdam |
|  | Learning Rate | 0.001 | 0.001 | 0.001 | 0.0005 | 0.0005 | 0.0007 |
|  | Temperature | 1.3 | 1.5 | 1.5 | 1.5 | 1.5 | 1.4 |
| GST-1.0 | Optimizer | Adam | RAdam | RAdam | RAdam | RAdam | RAdam |
|  | Learning Rate | 0.0005 | 0.0005 | 0.0007 | 0.0007 | 0.0007 | 0.0005 |
|  | Temperature | 1.0 | 0.5 | 0.5 | 0.5 | 0.7 | 0.5 |
| ReinMax | Optimizer | Adam | RAdam | RAdam | RAdam | RAdam | RAdam |
|  | Learning Rate | 0.0005 | 0.0005 | 0.0007 | 0.0007 | 0.0005 | 0.0005 |
|  | Temperature | 1.3 | 1.5 | 1.5 | 1.5 | 1.5 | 1.3 |
| ReinMax-Rao | Optimizer | Adam | RAdam | RAdam | RAdam | RAdam | RAdam |
|  | Learning Rate | 0.0005 | 0.0005 | 0.0007 | 0.0007 | 0.0007 | 0.0005 |
|  | Temperature | 1.0 | 0.7 | 1.0 | 1.0 | 1.0 | 0.7 |
| ReinMax-CV | Optimizer | Adam | RAdam | RAdam | RAdam | RAdam | RAdam |
|  | Learning Rate | 0.0005 | 0.0005 | 0.0005 | 0.0005 | 0.0005 | 0.0005 |
|  | Temperature | 1.0 | 1.3 | 1.1 | 0.7 | 0.7 | 0.7 |
|  | $\eta$ | 1.5 | 1.5 | 1.5 | 1.5 | 1.5 | 1.5 |

## F  Additional experimental results

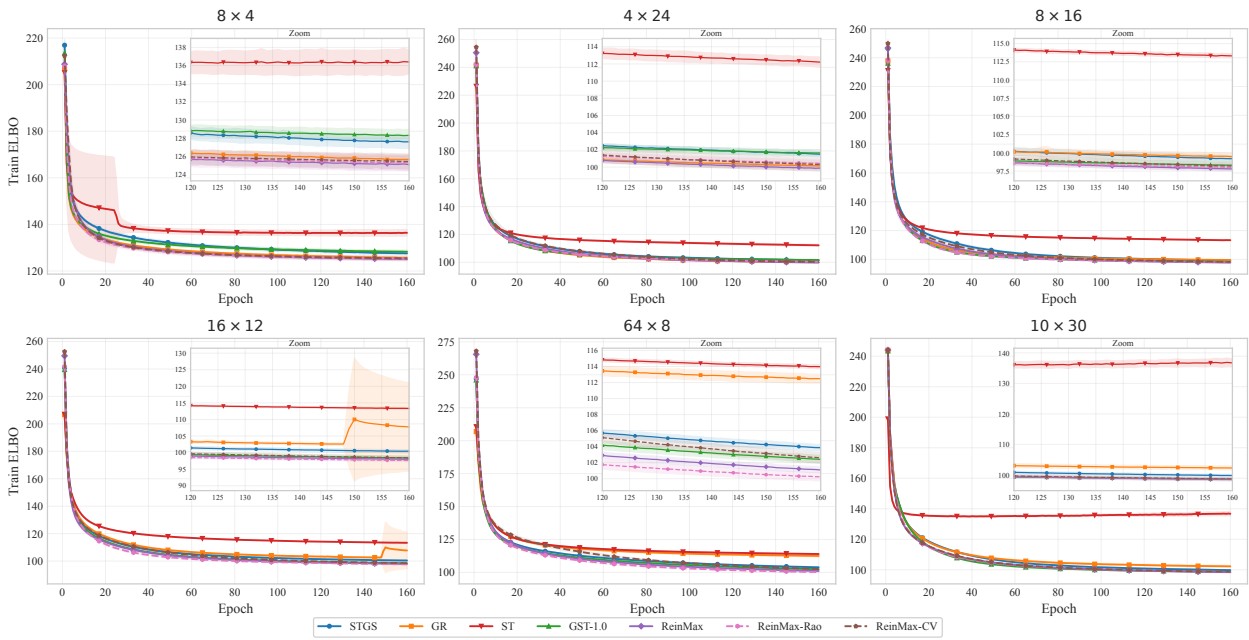

Figure 3: Negative train ELBO across training epochs for various methods

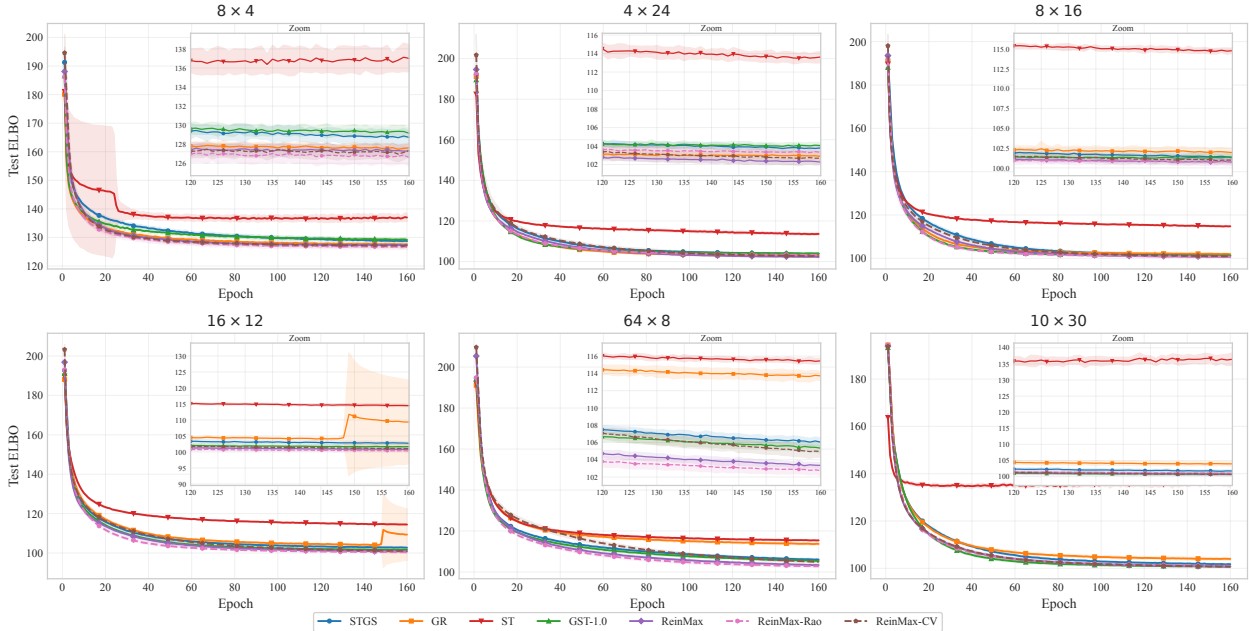

Figure 4: Negative test ELBO across training epochs for various methods

