# OpenReview forum: "Beyond ReinMax: Low-Variance Gradient Estimators for Discrete Latent Variables"
_TMLR — Accepted by TMLR_

### Review · Reviewer_wR5u · 2026-03-15

**Summary Of Contributions:**

The paper proposes two extensions to the straight-through gradient estimation approach ReinMax. The motivation is to reduce the variance of the estimator, leading to better performance of discrete VAEs. The main idea is to introduce the Gumbel-Softmax trick to the ReinMax estimator through a heuristic-based approach and control variates approach.

**Additional Comments:**

**Strengths**:
* Sufficient background on straight-through gradient estimation.
* Detailed theoretical justification.
* Extensive baseline comparison.
* Overall, it seems there are some performance gains over the baseline estimators.
* A promise for open-source implementation.


**Weaknesses**:
* Writing and language could be improved.
* The proposed estimates are more complex.
* No implementation details of the estimators.
* While the main contribution of the paper is theoretical, the empirical benchmark is simple and uses MNIST. My argument here is that due to the simplicity, it might be hard to obtain any conclusive or general insights regarding the applicability of the proposed estimators.

**Minor**:
* When referring to specific labels in the text,  “equation X” -> “Equation X”, “appendix” -> “Appendix”, “figure” -> “Figure”.

**Audience:**

Yes

**Audience Explanation:**

Yes, there is substantial research on gradient estimation and discrete VAEs.

**Claims And Evidence:**

Yes

**Claims Explanation:**

Yes, the claims are supported by clear evidence. See below for strengths.

**Requested Changes:**

* Eq. 7 - I think it requires further clarification: is it $Y_{\theta_j, D=I_i}$? What is the meaning of the lower $d$ above the equivalence?
* Computational cost: the authors report increased bias but lower variance; however, they do not discuss the computational cost of the proposed estimators compared to the baselines.
* Implementation: it seems that the implementation of the proposed estimators is more complex. I think a pseudocode or some practical implementation details would benefit the paper.
* See “Minor”.

---

> ### Author Response · Authors · 2026-04-13
> **Response to Reviewer wR5u**
>
> We thank the reviewer for the comments.
>
> re Eq. 7: The LHS is the $j$-th entry of $Y_{\theta, D}$ which is the random variable $(\theta + G)$ conditioned on $D=I_i$. The $d$ denotes that it is equivalent in distribution to the RHS.
>
> re computation cost: We have added a comparison to Appendix D. Our (unoptimised) implementation of the proposed approaches has a small overhead compared to Gumbel-Rao.
>
> re implementation: We have included the pseudo-code in Appendix B. In addition, we will release the code upon acceptance.
>
> re “Minor”: Thanks for pointing these out. We have fixed them.
>
> Please let us know if you have any further questions/requests.

---

### Review · Reviewer_FBKH · 2026-03-24

**Summary Of Contributions:**

This paper aims to reduce the variance of ReinMax, a gradient estimation method for machine learning models with discrete latent variables. The authors propose two estimators: ReinMax-Rao, which incorporates the Rao-Blackwellisation technique, and ReinMax-CV, which employs control variates.

**Audience:**

No

**Audience Explanation:**

- While machine learning involving discrete variables is an important topic in the community, the transformation from ReinMax to the proposed methods does not include any technical innovations, providing few takeaway messages from this work.

- The experiments provide evaluations of the proposed methods only in limited toy scenarios. Readers cannot judge whether the proposed methods lead to meaningful improvements in modern deep learning cases. For instance, evaluations on larger-scale datasets, more complex architectures, or applications beyond VAEs (such as discrete structure learning or reinforcement learning) are necessary.

- I do not understand the purpose and academic value of Section 5. From my perspective, the authors simply present many formulas without clear main messages. The contribution of this entire section remains unclear.

**Claims And Evidence:**

No

**Claims Explanation:**

- Firstly, the title "Beyond ReinMax" overclaims their contributions. The proposed methods simply apply the Gumbel-Rao estimator and control variates to ReinMax in a very straightforward manner, without introducing novel theoretical insights or methodological innovations.

- I fully disagree with their claim that "Extensive experiments ... show that". The proposed methods are only evaluated by ELBO of VAEs trained in very limited scenarios (e.g., very simple MLP architectures, dataset limited to MNIST). It is very difficult to judge whether the performance gain is meaningful even in these limited experiments.

- While ReinMax-Rao and ReinMax-CV achieve lower gradient variance compared to ReinMax during training, their variances remain larger than those of most other baselines such as Straight-Through and Gumbel-Rao (see Figure 1, right panel). If variance reduction is claimed as the main contribution, this point requires more detailed discussion.

**Requested Changes:**

- Please refer to my comments above for major revision of this work.

- Section 2 introduces too many concepts, and it is hard to understand the authors' intention for including them. This section should be much more concise and provide clear guidance for readers.

- There are too many complicated notations, which make reading the paper very challenging. Thorough rewriting is needed for the entire manuscript.

- The temperature setting in Eq (18) is not well explained. Why are the temperatures in the first and last terms set to 1 without tuning unlikely to the others? The theoretical rationale for this choice should be provided.

- Some typos in the formulations should be fixed for easier reading:
  - Eq (17) mistakenly includes $G$ in the first term (should be $D$).
  - $p(x)$ on the leftmost side of Eq (19) is not defined (presumably should be $p_{\psi}(x)$).

- The authors mention that "ReinMax-CV has higher bias than ReinMax in practice." Are there any experiments supporting this claim?

- A comparison of computational costs is lacking. While it is stated that methods using the Gumbel-Rao approximation are "generally around three times slower," actual computation time comparisons should be provided.

---

> ### Author Response · Authors · 2026-04-14
> **Response to Reviewer FBKH**
>
> We thank the reviewer for the comments.
>
> re Section 2 introducing too many concepts and complicated notations: We respectfully disagree and kindly ask the reviewer to have another look at the submission. Reviewer FL2d found that “The related work is nicely put together and enjoyable to read; it connects many of the different approaches concisely and clearly” and “the rewrite of ReinMax as ( ) introduces a new perspective”, and Reviewer wR5u mentioned “Sufficient background on straight-through gradient estimation”.
>
> re temperature setting in eq (18): The first and last terms are from ReinMax, where the last term always has $\tau=1$. Thus only the first three terms should be considered for tuning. However, using the same temperature for the first three terms is not ideal as Liu et. al. (2023)  found that ReinMax (first term) prefers higher temperatures while Gumbel-Softmax (second term) and Gumbel-Rao (third term)  prefer lower temperatures. We previously tried setting the temperature of the first term to the best values used by ReinMax and only tuning the second and third terms, and found that setting $\tau=1$ for the first term as described in the paper yielded slightly better performance. Secondly, the key theoretical property of ReinMax is that in expectation it is equal to the second-order approximation  (equation 13) which only holds when $\tau=1$. Setting a higher temperature has the same effect as our methods (reduced variance but increased bias) so it is not necessary to tune the temperature for the first term.
>
> re bias: This is shown in Figure 1. We measure the bias using the cosine similarity between the gradient estimates and the exact gradient (see Appendix C for more details). ReinMax-CV has lower cosine similarity and hence higher bias than ReinMax.
>
> re comparison of computation cost: We have now provided this in Appendix D.
>
> Please let us know if you have any further questions/requests.

---

### Review · Reviewer_FL2d · 2026-03-25

**Summary Of Contributions:**

This paper analyzes the variance behavior of the ReinMax gradient estimator and proposes two modifications, ReinMax-Rao and ReinMax-CV, which use Rao–Blackwellization and control variates to reduce variance. The authors show that ReinMax can be rewritten as a combination of two straight-through (ST) estimators and argue that one of these terms introduces additional variance because it depends on the sampled discrete variable. The proposed estimators replace this noisy term with lower-variance alternatives (by integrating existing methods, such as Gumbel-Rao). Experiments on discrete VAEs show slightly reduced gradient variance and improved ELBO compared to several baselines. Overall, the analysis is clear and the proposed estimators are reasonable extensions of ReinMax, though the contribution feels somewhat incremental and the experiments are rather limited.

**Audience:**

Yes

**Audience Explanation:**

This paper provides a useful analysis of the ReinMax estimator and proposes reasonable variance-reduction variants. The paper is well written and concise. However, the contribution is fairly incremental and the experimental section could be strengthened. I would lean toward acceptance if the authors address the concerns above.

**Claims And Evidence:**

Yes

**Claims Explanation:**

## Strengths
- The related work is nicely put together and enjoyable to read; it connects many of the different approaches concisely and clearly.
- The rewrite of ReinMax as  $2 \text{ST}(D, \theta_D) − \frac{1}{2} \text{ST}(D, \theta)$  introduces a new perspective. However, it might be worth explaining the intuition for where the variance comes from in more detail. E.g., since the Jacobian is evaluated closer to the simplex boundary, or perhaps it can be seen by inspecting the terms introduced after applying the chain rule to the gradient.
- Experiments measure both gradient variance and final ELBO performance, which helps support the claims.
- The intuition showing how $\beta=1/2$ relates to trapezoidal integration is interesting.

## Weaknesses / Suggestions

- The contribution feels somewhat incremental relative to Liu et al. (2023). The setup, notation, and experiments are quite similar, and the proposed methods mainly apply standard variance-reduction ideas (Rao–Blackwellization and control variates).
- It would help to explain how these estimators relate to practical implementations using surrogate gradients (e.g., `stop_gradient(D - p) + p`). The paper discusses the estimators analytically but does not connect them clearly to how they are implemented in autodiff frameworks.
- Some claims in the introduction lack citations (e.g., statements about Gumbel-Rao being three times slower).
- Important related work is missing. In particular, *Discrete Variational Autoencoding via Policy Search (DAPS, Drolet et al.)* shows that likelihood-ratio / REINFORCE-style estimators can scale to large discrete latent spaces (e.g., ImageNet-scale models). The introduction of this paper suggests that likelihood-ratio methods are impractical in modern deep learning settings, which appears inconsistent with these recent results. The authors should cite and discuss this line of work and potentially soften the claim about the impracticality of score-function estimators.
- Equations for how the reported bias and variance are computed (in the plots) might be nice to have in the appendix. In particular, the equation for computing the exact gradient in practice is a bit unclear-- I assume this was computed by enumerating over all possible latent codes, but clarifying this would be appreciated.
- The supplementary material for section 5 is a bit lacking (having only figure 2). It would be very helpful to add a graphical figure that displays the geometric intuition of the integration (i.e., the effect of beta on the shape). Furthermore, would it be possible to run a small ablation on the degree 3 polynomial integration method (if the setup is simplified)? Since the motivation of section 5 is to explore the different integration methods, it feels open-ended to leave it out.
- Training curves are missing and number of seeds per experiment are missing. Showing optimization trajectories would help illustrate convergence behavior.
- Minor issue: the paper reports negative ELBO but refers to it simply as ELBO.
- Minor notation issue: in Section 2.1 the derivative appears as $\frac{d \pi}{d D}$, which seems reversed relative to the straight-through chain rule.
- Minor comment: The choices of latent bottleneck configurations seem somewhat arbitrary. It might be clearer to control for information capacity (e.g., 32, 64, 128 bits) as done in several recent papers (e.g. FSQ and DAPS), despite being a different setup than Liu et al 2023.

**Requested Changes:**

Please see the Weaknesses / Suggestions from above.

---

> ### Author Response · Authors · 2026-04-14
> **Response to Reviewer FL2d**
>
> We thank the reviewer for the comments/suggestions.
>
> re implementation details: Our implementation is based on the one provided by Liu et. al. (2023) which does not use surrogate gradients but instead obtains the upstream gradient via autograd and implements the $\frac{dsoftmax(\theta)}{d\theta}$terms explicitly. Please see Appendix B for pseudocode.
>
> re run-time claim: We have added a comparison in Appendix D.
>
> re related work: Thanks for pointing this out. We have added the suggested reference in the introduction.
>
> re equations for biases and variances: Yes, the exact gradients are computed by enumerating over all possible latent codes. We have provided the details in Appendix C.
>
> re seeds and training curves: We use 10 seeds for each experiment. We have now included this number in Appendix E, as well as the training curves in Appendix F.
>
> re negative ELBO: Thanks for pointing this out. We will fix this in the paper.

---

### Decision · Action_Editor_9xQ2 · 2026-05-12

**Recommendation:** Accept as is

**Audience:**

Yes

**Audience Explanation:**

Discrete variational autoencoding is a relevant problem, with potential applications for transformers and similar architectures. While the resulting methods do not seem to achieve state-of-the-art performance, the insights on variance reduction are of potential interest for some of TMLR's audience

**Claims And Evidence:**

Yes

**Claims Explanation:**

The paper investigates variance reduction for the ReinMax gradient for discrete variational autoencoding. It proposes two variants based on existing techniques (Gumbel-Rao and control variates). The resulting variants are evaluated on MNIST and compared with ReinMax  and related methods in terms of ELBO, as well as estimates of the bias and variance.

The novelty of the method is modest, and the evaluation does not include SOTA methods for discrete auto encoding or more challenging datasets. However, the paper does provide novel insights on variance reduction for ReinMax and provides sufficient evidence to support these claims. As there don't seem to be any factual wrong or overly bold statements, I argue that the acceptance threshold for TMLR are met.